# An Intranuclear *Sodalis*-Like Symbiont and *Spiroplasma* Coinfect the Carrot Psyllid, *Bactericera trigonica* (Hemiptera, Psylloidea)

**DOI:** 10.3390/microorganisms8050692

**Published:** 2020-05-08

**Authors:** Saptarshi Ghosh, Noa Sela, Svetlana Kontsedalov, Galina Lebedev, Lee R. Haines, Murad Ghanim

**Affiliations:** 1Department of Entomology, Institute of Plant Protection, Volcani Center, ARO, HaMaccabim Road 68, P.O. Box 15159, Rishon LeZion 7528809, Israel; sunnysaptarshi@gmail.com (S.G.); nasvetla@yahoo.com (S.K.); galinal@volcani.agri.gov.il (G.L.); 2Department of Plant Pathology and Weed Research, Institute of Plant Protection, Volcani Center, ARO, HaMaccabim Road 68, P.O. Box 15159, Rishon LeZion 7528809, Israel; noa@volcani.agri.gov.il; 3Department of Vector Biology, Liverpool School of Tropical Medicine, Pembroke Place, Liverpool L3 5QA, UK; Lee.Haines@lstmed.ac.uk

**Keywords:** psyllid, *Bactericera trigonica*, *Candidatus* Liberibacter solanacearum, secondary endosymbiont, *Sodalis*, *Spiroplasma*, intranuclear bacteria

## Abstract

Endosymbionts harbored inside insects play critical roles in the biology of their insect host and can influence the transmission of pathogens by insect vectors. *Bactericera*
*trigonica* infests umbelliferous plants and transmits the bacterial plant pathogen *Candidatus* Liberibacter solanacearum (CLso), causing carrot yellows disease. To characterize the bacterial diversity of *B. trigonica*, as a first step, we used PCR-restriction fragment length polymorphism (PCR-RFLP) and denaturing gradient gel electrophoresis (DGGE) analyses of 16S rDNA to identify *Sodalis* and *Spiroplasma* endosymbionts. The prevalence of both symbionts in field-collected psyllid populations was determined: *Sodalis* was detected in 100% of field populations, while *Spiroplasma* was present in 82.5% of individuals. Phylogenetic analysis using 16S rDNA revealed that *Sodalis* infecting *B. trigonica* was more closely related to symbionts infecting weevils, stink bugs and tsetse flies than to those from psyllid species. Using fluorescent in situ hybridization and immunostaining, *Sodalis* was found to be localized inside the nuclei of the midgut cells and bacteriocytes. *Spiroplasma* was restricted to the cytoplasm of the midgut cells. We further show that a recently reported *Bactericera trigonica* densovirus (BtDNV), a densovirus infecting *B. trigonica* was detected in 100% of psyllids and has reduced titers inside CLso-infected psyllids by more than two-fold compared to CLso uninfected psyllids. The findings of this study will help to increase our understanding of psyllid–endosymbiont interactions.

## 1. Introduction

Symbiotic bacterial associations with insect hosts can be either obligate or facultative in nature. Obligate bacterial symbionts are always vertically transmitted, are mostly located in a specialized structure called the bacteriome and typically co-speciate with their insect hosts [1,2]. Obligate symbionts share metabolic pathways with their hosts and synthesize essential amino acids and vitamins [3]. Facultative bacterial symbionts have been more recently acquired by their insect hosts, colonize multiple insect tissues and organelles and can be transmitted either vertically or horizontally [2,4]. Facultative symbionts provide conditional benefits to their hosts such as heat tolerance [5,6], defense against pathogens [7,8] and parasites [9], resistance to pesticides [10], and reproductive advantages such as increased fecundity and adult emergence [11]. Facultative symbionts can also influence the insect’s ability to transmit insect-borne pathogens [12,13].

Psyllids (Hemiptera, Psylloidea), also known as plant lice, are tiny, phloem-sucking, plant-specific insects of enormous agricultural importance as they transmit devastating diseases to food crops. A gamma-proteobacterial symbiont, *Candidatus* Carsonella ruddii, is harbored inside the bacteriocytes of all psyllid species [14,15]. Alongside *Candidatus* Carsonella, a diverse group of facultative symbionts such as *Arsenophonus* and *Sodalis* (gamma-proteobacteria), as well as *Rickettsia* and *Wolbachia* (alpha-proteobacteria) contribute to the psyllid endosymbiotic microflora [14,16,17,18]. *Bactericera trigonica* (*B. trigonica*) is a psyllid species infesting umbelliferous crops in the Mediterranean and Middle East. *B. trigonica* transmits the emerging, phloem-restricted plant pathogen *Candidatus* Liberibacter solanacearum (CLso) (haplotype D), which causes significant economic losses to crops such as carrots, celery and parsley [19,20,21,22]. Carrots are a major vegetable export from Israel to European markets; however, since 2009, an epidemic of CLso transmitted by *B. trigonica* has caused significant financial shortfalls for those in the carrot industry in Israel and Europe. Over 80% of the carrot plants with yellow symptoms and 10% of asymptomatic ones tested positive for CLso in Israel in 2015 [23].

CLso is transmitted by *B. trigonica* in a circulative, persistent manner; ingested bacteria migrate across insect midgut barriers to reach the hemolymph and then penetrate the salivary glands before the insect can inoculate the plant phloem upon feeding [21,22]. Several insect-borne plant pathogens with tissue-specific lifecycles can interact with endosymbionts to influence transmission efficiency. For example, endosymbionts of the whitefly *Bemisia tabaci* and of several aphid species enhance the transmission of pathogenic, insect-borne plant viruses by protecting the viruses from the hostile environment (up-regulated immune factors) in the host hemolymph [13,24]. In contrast, the presence of a symbiont, *Wolbachia spp.*, reduces the capacity of mosquitoes to transmit Dengue virus, which clearly demonstrates how these “defensive” endosymbionts can be exploited to control virus transmission [25,26].

The characterization of the endosymbiotic diversity and their niche inside *B. trigonica* is a first step to understanding their direct or indirect interactions with CLso. In this study, we investigated the bacterial endosymbionts inside *B. trigonica* and identified two facultative symbionts, *Spiroplasma sp.* and *Sodalis*-like bacteria that colonize *B. trigonica* alongside *Carsonella sp.*, the primary symbiont. Additionally, we also describe the prevalence and dynamics of the recently reported *Bactericera trigonica* densovirus (BtDNV) [27] inside the psyllids with and without CLso. We also localize the symbionts within insect tissues to understand their spatial distribution and possible influence on host physiology. Moreover, we show that psyllids infected with CLso have reduced titers of BtDNV compared to psyllids free of CLso.

## 2. Materials and Methods

### 2.1. Insect Collection and Culture

*B. trigonica*—CLso-infected and uninfected populations—were collected from carrot fields in Southern Israel during the summer of 2015. The CLso infection status (uninfected/infected) of collected psyllid populations was determined by the qPCR (Table 1) of celery plants used to rear the psyllids after 2 months from the release of insects. Laboratory CLso (haplotype D)-infected (100% infection, reared on CLso infected plants) and uninfected populations (reared on uninfected plants) were maintained in separate cages on organically raised celery (*Apium graveolens*, variety Spartacus, which is a suitable host for both CLso and *B. trigonica*) under controlled environmental conditions (25 °C, 60% R.H, 14 hL: 10 hD). The presence and absence of CLso from psyllid populations were routinely confirmed every 3 months by performing qPCR on insects and the celery plants used for rearing.

### 2.2. PCR-RFLP and Denaturing Gradient Gel Electrophoresis (DGGE) of 16S rDNA 

Total insect DNA was extracted from surface-sterilized (immersion in 70% ethanol for 5 min) individual CLso-infected and uninfected insects by the CTAB extraction method [28]. Briefly, each insect was homogenized in 500 µL of CTAB buffer containing 1 µL of β-mercaptoethanol and then incubated at 37 °C for 45 min. Equal volumes of phenol:chloroform:isoamyl alcohol (25:24:1) were added, and then samples were centrifuged at 1300 rpm for 10 min. To the supernatant, equal volumes of isopropanol were added, and the DNA was allowed to precipitate at −20 °C for 1 h. DNA pellets were obtained by centrifugation at 1300 rpm for 10 min followed by washing with 70 % ethanol. The pellets were air-dried and dissolved in 50 µL DNase free water.

A 1.5 kb fragment of the eubacterial 16S rDNA was amplified by using the universal eubacterial primers fD1 (5′ AGAGTTTGATCCTGGCTCAG 3′) and rP1 (5′ GGTTACCTTGTTACGACTT 3′) [29] with thermal cycling conditions of 95 °C for 3 min, followed by 35 cycles of 95 °C for 20 s, 52 °C for 30 s and 72 °C for 2 min. The PCR products were gel-purified and cloned into pGEM-T easy vector (Promega, U.S.A). Clones containing the insert were confirmed by colony PCR with T7F and SP6 promoter primers. Colony PCR products (300 colonies) were digested with *Taq* I (ThermoFisher Scientific) and visualized on 2% agarose gels. Five clones each for unique restriction fragment length polymorphism (RFLP) profile patterns were sequenced (HyLabs, Rehovot, Israel) for both directions.

Alternatively, a 500 bp fragment of the eubacterial 16S rDNA was PCR amplified using primer set 341F with GC clamp (5′CGCCCGCCGCGCCCCGCGCCCGTCCCGCCGCCCCCGCCCGCCTACGGGAGGCAGCAG 3′) and 907R (5′ CCGTCAATTCMTTTGAGTTT 3′) [30] followed by denaturing gradient gel electrophoresis (DGGE) on a vertical gel containing 8% polyacrylamide with a linear gradient of denaturants increasing from 0% to 60%. Electrophoresis was performed in 1× TAE at 200 V for 4 h at 60 °C. Gels were stained with ethidium bromide and photographed on a UV trans-illuminator. The DGGE bands were excised, purified and sequenced (HyLabs, Rehovot, Israel).

### 2.3. Prevalence of Symbionts in Field Collected B. trigonica 

Forty psyllid samples (20 males and 20 females) collected from different carrot fields in the south/north of Israel (Saad, Khavat Eden) were tested individually for the presence of secondary symbionts and a densovirus infecting *B. trigonica* (BtDNV) [27] using specific primers (Table 1). A primer amplifying the actin gene of *B. trigonica* (designed using actin sequences in Genbank, KT185024.1 and XM_008470468.2) was used as an internal control (Table 1). 

### 2.4. Relative Quantification of Symbionts in CLso-Infected/Uninfected Psyllids 

Relative amounts of *Sodalis*, *Spiroplasma* and BtDNV quantified by qPCR was compared between adult psyllids reared on CLso-infected and uninfected celery plants. *Sodalis*, *Spiroplasma* and BtDNV infected 100% of the laboratory psyllid population used in this study. Total DNA was extracted from a pool of five insects (three female and two male, 5–10 days old) by the CTAB extraction method as described previously. Primer sets were designed to amplify specific targets of *Sodalis*, *Spiroplasma* and BtDNV (Table 1). Nine replicates of CLso-infected and uninfected samples each were used for the relative quantification of *Sodalis* and Spiroplasma, while 14 replicates each were used for BtDNV. A primer amplifying the actin gene of *B. trigonica* was used as an internal reference to normalize the symbiont titres. The efficiency of all primers used ranged between 95%–103%. The PCR reactions were run in 15 µL reactions with 7.5 µL of Thermo Scientific Absolute Blue qPCR SYBR Green ROX Mix (Thermo Scientific) and 300 nM of forward and reverse primer each. The cycling conditions were as follows: 95 °C for 10 min; 40 cycles of 95 °C for 10 s, 58 °C for 20 s and 72 °C for 30 s; and a melt curve of 60–90 °C with a holding step for one second for every 0.5 °C rise. The relative bacteria quantities relative to the actin gene of *B. trigonica* were calculated using the Delta Ct method. Mean quantities of the symbionts were analyzed using a simple linear model with log_e_-transformed data, and statistical inference was based on the results of a one-way analysis of variance (ANOVA). 

### 2.5. Fluorescent in Situ Hybridization (FISH) and Immunostaining for Bacteria Localization

Insect internal organs (guts, ovary, fat bodies, bacteriome, testes and salivary glands) from CLso-infected and CLso-free *B. trigonica* were dissected into 1× phosphate saline buffer (pH 7.2) under a stereomicroscope. Bacteria (CLso, *Sodalis* and *Spiroplasma*) were localized in the dissected organs by fluorescent in situ hybridization (FISH) as described in Ghanim et al. (2016) [32]. Briefly, the dissected organs were fixed in Carnoy’s fixative (chloroform: ethyl alcohol: acetic acid, 6:3:1, *v*/*v*) followed by overnight hybridization with the probes at room temperature.

### 2.6. Immunolocalization of Sodalis and CLso 

*Sodalis* was immunolocalized in *B. trigonica* using the mouse anti-*Sodalis glossinidius* GroEL monoclonal antibody [33] (a gift from Prof. Terry Pearson, University of Victoria, Victoria, BC, Canada). Midguts from adult psyllids were dissected in 1× PBS buffer followed by the immunolocalization of *Sodalis* using anti-*Sodalis* GroEL primary monoclonal antibody (1:100) as described in Ghanim et al. (2017) [21]. 

### 2.7. Sequencing and Genome Assembly of the Sodalis-Like Symbiont

Total nucleic acids were extracted from 20 surface-sterilized CLso-infected adult psyllids by the CTAB method, as described previously. Genomic DNA was sequenced using Illumina HiSeq 2000 system and paired-end libraries, with an average insert size of 300 bp, were constructed (BGI, Hong Kong). A total of 61,673,620 reads were obtained from the pair-ended libraries with a final yield of 61,658,314 million (99.98%) clean reads after the trimming of adaptor and low-quality reads using the trimmomatic software [34]. Pair-end library reads were assembled using the pipeline a5 assembly [35] and 625,635 contigs were generated with an N50 of 1198 bp. The assembly was then integrated into the Contig Assembly of Prokaryotic Draft Genomes Using Rearrangements (Available online: http://genome.cs.nthu.edu.tw/CAR/) pipeline [36] with the *Sodalis glossinidius* genome (GCF_000010085.1) as a reference.

The *Sodalis*-like bacterial genome associated with *B. trigonica* contains 753 contigs. This genome was uploaded to the RAST server for annotation [37] and compared with other *Sodalis* genomes available in public repositories. Phylogenetic analysis was performed by concatenating all homologous proteins with the aid of the software OrthoFinder version 2.1.2 [38]. The Multiple Sequence Alignment (msa) algorithm from Orthfinder was used to create the phylogenetic trees with 100 bootstraps in phyml software version 3.0 (Available online: http://www.atgc-montpellier.fr/phyml/) [39]. The genome of the *Sodalis* endosymbiont of *B. trigonica* was uploaded to the NCBI database under the accession GCA_003668825.1.

## 3. Results

### 3.1. Identification and Prevalence of Symbionts Associated with B. Trigonica

The Sanger sequencing of the 16S rDNA clones with different RFLP profiles and the bands obtained by DGGE analyses identified *Sodalis-like* sp., *Spiroplasma* sp. and CLso by comparing the sequences with other bacterial sequences using the NCBI BLAST algorithm. *Sodalis*-like bacteria have been previously identified from other psyllids: the eucalyptus psyllid (*Blastopsylla occidentalis*), potato psyllid (*Bactericera cockerelli*) and alder psyllid (*Psylla floccosa*) (Figure 1). 

The phylogenetic analysis of the 16S rDNA nucleotide sequences clustered *Sodalis* infecting *B. trigonica* with *Sodalis* species found in weevils, stinkbugs, tsetse flies and a human wound (with > 98% similarity), but not with the *Sodalis* spp., which was previously identified from other psyllids or other members of the Sternorrhyncha (aphids, mealybugs) (Figure 1). The phylogenetic analysis of the 16S rDNA fragment grouped the *Spiroplasma* infecting *B. trigonica* closely with *Spiroplasma ixodetis* Y32 strain (99.3% nucleotide identity)—a symbiont prevalent in ticks, mosquitoes, pea aphids and ladybird beetles (Figure 2). 

The prevalence of the identified symbionts in the *B. trigonica* were analyzed by PCR. *Sodalis* was prevalent in 100% of the field *B. trigonica* samples tested (40/40), while *Spiroplasma* was detected in 82.5% (33/40) of the insects tested. We also tested the prevalence of a previously reported densovirus (BtDNV) found to infect *B. trigonica* in this study. Densovirus (BtDNV) was also detected in 100% of the *B. trigonica* samples tested (40/40). 

### 3.2. Relative Quantification of Symbionts in CLso-Infected/Uninfected Psyllids

To test whether CLso, the bacterial plant pathogen transmitted by *B. trigonica*, directly interacts with the identified endosymbionts, we quantified the relative amounts of *Sodalis*, *Spiroplasma* and BtDNV in psyllids infected and uninfected with CLso. Relative amounts of the bacterial symbiont, *Sodalis* (Figure 3A) and *Spiroplasma* (Figure 3B) did not differ significantly between psyllids infected with or free of CLso. However, psyllids infected with CLso had significantly reduced quantities of BtDNV by greater than two-fold (F = 26.84, *p* < 0.001) compared to psyllids free of CLso (Figure 3C). 

### 3.3. Tissue Localization of Sodalis, Spiroplasma and CLso within B. trigonica

Using FISH, *Sodalis* DNA was localized inside the nuclei and nucleolus as well as in the cytoplasm of midgut cells (Figure 4A–F) and both inside and outside the nuclei in bacteriocyte cells (Figure 5A–H). *Sodalis* was not detected in the ovaries/testes or salivary glands (not shown). The immunolocalization of *Sodalis* using mouse anti-*Sodalis glossinidius* GroEL monoclonal antibody also confirmed its endonuclear location inside midgut cells (Figure 6A,B). Attempts were made to examine if CLso co-localized with *Sodalis*, which may imply *Sodalis*–CLso interactions within *B. trigonica*. As evidenced by Appendix A, the bacteria did not co-localize; CLso localized in a predicted stripe-like pattern within the midgut cells (Appendix A), as observed previously in our work with *Candidatus* Liberibacter asiaticus (CLas) in the Asian citrus psyllid (ACP) *Diaphorina citri* (16, 28). 

*Spiroplasma* localization was restricted to the midgut cell cytoplasm (Figure 7A,B,E,F) using FISH, and DNA was not detected in bacteriocytes (Figure 7G,H). *Spiroplasma* appeared to aggregate in patches around the nucleus using a specific fluorescent probe and under UV light to visualize the DAPI staining of the nuclei. The *Spiroplasma* patches seemed to segregate with cell division and were included in the dividing cells (Figure 7A,B,E,F). The co-localization of *Spiroplasma* with CLso, as seen in Figure 7A–D,F, was not observed. CLso showed the same stripe-like patterns as before but did not co-localize with *Spiroplasma* in the cell cytoplasm.

### 3.4. Sequence of the Sodalis-Like Endosymbiont of Bactericera trigonica

Using the Illumina assembly, we produced a preliminary draft sequence for *Sodalis*, which was used to characterize the basic genomic properties and metabolic functions of this bacterium. The sequence length of the draft assembly of *Sodalis* was 1.58 Mb with 55.7% GC content. The draft genome consisted of 746 scaffolds with an N50 of 2843 bp. Phylogenetic analysis using orthologous genes from all available *Sodalis* genome sequences re-clustered the *Sodalis* endosymbiont from *B. trigonica* with that of *Sodalis glossinidius,* the secondary symbiont found in tsetse flies (Figure 8). However, the *B. trigonica*-associated *Sodalis* genome size was reduced by more than 2.7 times compared to *Sodalis glossinidius* (4.29 Mb), and the former lacks most orthologous genes across functional categories such as biosynthetic pathways for most amino acids, vitamin biosynthesis and carbohydrate utilization.

## 4. Discussion

Symbiont diversity and localization inside their insect hosts have immense ecological importance, with potential downstream applications on vector control. In this study, we characterized two endosymbiotic bacteria that co-habit field populations of the carrot psyllid *Bactericera trigonica*. *Sodalis sp* are Gram-negative bacterial endosymbionts commonly associated with many insects such as tsetse flies [40], weevils [41], stink bugs [42], beetles [43], aphids [44], louse flies [45] and even some psyllid species [14,17]. Although *Sodalis* has been previously detected in other psyllid species [17], such as the potato psyllid (*Bactericera cockerelli*), eucalyptus psyllid (*Blastopsylla occidentalis*) and the alder psyllid (*Psylla floccosa*), the infection prevalence of *Sodalis* in these insect species vary considerably. *Sodalis* was detected in 100% of the *B. trigonica* samples tested in Israel. We expected that *Sodalis* infecting *B. trigonica* would be closely related to *Sodalis*-like species that infect other psyllid hosts. However, to our surprise, the phylogenetic analysis of the 16s rDNA nucleotide sequence revealed the highest similarity to *Sodalis* sp., infecting stink bugs and weevils but not with those infecting other closely related hemipterans such as psyllids, aphids and mealybugs. The 16S rDNA sequence of *Sodalis* infecting *B. trigonica* was greater than 98%, which was similar to that of *Sodalis praecaptivus* (a pathogenic isolate from a human wound) and *Sodalis glossinidius*, a secondary symbiont in tsetse flies. It is possible that the *Sodalis* associated with *B. trigonica* has been recently acquired from *Sodalis*-like symbionts that colonize other insect species via horizontal transfer. Since *S. glossinidius* [40,46] and *S. praecaptivus* [47] can be cultured in vitro, it has been suggested that these *Sodalis* strains may contribute to an environmental pool for symbiont acquisition [41]. 

*Spiroplasma* was found to infect 82.5% of *B. trigonica* and is closely related to *Spiroplasma ixodetis*. How *Spiroplasma* was acquired by *B. trigonica* remains unknown, but this is maintained in psyllids through vertical transfer from parents to offspring. *Spiroplasma* species closely related to *S. ixodetis* are known to induce male killing in butterfly and ladybird beetle larvae [48]. It remains unknown whether the strain of *Spiroplasma* that infects *B. trigonica* can produce a similar phenotype. However, the exclusive localization of *Spiroplasma* in the adult midgut tissue of *B. trigonica* limits its potential for reproductive manipulation. *Wolbachia* and *Arsenophonus*, the other two predominant secondary symbiont species known to infect other psyllid species, were not detected in *B. trigonica* collected from our field sites. Moreover, in this study, we also estimated the prevalence of recently described densovirus (BtDNV) infecting *B. trigonica* [27]. BtDNV was detected in 100% of the psyllid samples tested. Interestingly, adult psyllids infected with CLso had significantly reduced titers of BtDNV compared to CLso-uninfected psyllids. Densovirus titers inside insects have been previously shown to be influenced by the presence of viruses transmitted by the insect [49] or other symbionts [50]. The reasons for reduced titers of BtDNV inside CLso-infected psyllids remain unknown but could be due to competitive interactions between BtDNV and CLso inside the psyllid. 

*Sodalis* inside *B. trigonica* occupy an unusual intranuclear niche within midgut cells, as confirmed by FISH and immunostaining using a *Sodalis*-specific monoclonal antibody. However, we unsuccessfully attempted to visualize the intranuclear habitat of *Sodalis* inside the midgut cells by TEM. The possible reasons for this failure possibly included the low titers of *Sodalis,* which make it difficult to visualize *Sodalis* inside the densely stained nuclei of the midgut cells. Intranuclear symbiosis is a rare phenomenon, and only *Rickettsia* and *Orientia*-like organisms have been reported to inhabit to insect cell nuclear organelles [51]. Hemipteran insects harbor *Rickettsia*-like bacteria inside Malpighian tubule nuclei, midgut cells, salivary glands and ovaries [52,53,54]. This is the first time a species of *Sodalis* has been reported to colonize the nuclei of insect cells, and the evolutionary reason for such compartmentalization is intriguing. Intranuclear location might confer advantages to symbionts such as protection from cytoplasmic defense mechanisms. The nucleus also provides a rich pool of proteins and nucleic acid resources for symbiont growth [51]. Eukaryotic nuclei control gene regulation and cell division. Thus, intracellular pathogenic bacteria can selectively target the host cell nucleus to subvert host defenses by releasing effector molecules, which enter the host cell nucleus and hijack nuclear processes [55]. Thus, the presence of *Sodalis* inside the midgut cell nuclei allows room for the direct manipulation of the host cell by targeting important nuclear processes including CLso establishment. FISH analyses of infected nuclei did not exhibit any co-localization of *Sodalis* with *Spiroplasma* or CLso in the psyllid midguts, which suggests that any influence these two microorganisms have on the host–pathogen interactions will likely be indirect. *Sodalis* sp. can facilitate pathogen establishment in insect vectors; for example, it has been proposed that *Sodalis glossinidius* synthesizes exochitinases, which degrade chitin and produce lectin-inhibitory sugars. These sugars may help trypanosomatid parasites establish in the tsetse fly (vector) midgut by inhibiting the production of trypanocidal midgut lectins [40,56]. Whether *Sodalis* inside *B. trigonica* also conditions the psyllid for CLso acquisition in similar ways remains unknown; however, a complete pathway for β-galactosidase synthesis remains intact. Unfortunately, the functional activity of *Sodalis* on CLso establishment could not be tested in this study due to the limitation of acquiring *Sodalis-*free psyllids as 100% of the field isolates contained this symbiont.

Paratransgenesis is a vector control technology that involves the manipulation of an insect phenotype by expression from a transformed symbiont [57]. Both *Sodalis* and BtDNV are good candidates for paratransgenesis due to their multiple-insect host range, efficient vertical transmission to offspring and amenability to genetic transformation. Recombinant *Sodalis glossinidius* was engineered to express anti-trypanosomal nano-antibodies to limit trypanosome development in tsetse midguts [58,59]. The in vivo proximity of *Sodalis* to CLso within the psyllid midgut and its localization in the nucleus are two attributes that increase its potential for delivering foreign DNA into the host cell or constitutively expressing dsRNA to disrupt CLso establishment and transmission. The successful, robust in vitro culture of *Sodalis* from *B. trigonica* and the re-introduction of a recombinant *Sodalis* into the psyllid are obstacles which are yet to be overcome before we can constitutively express dsRNA to disrupt CLso establishment in the insect gut. However, the ability to culture *Sodalis ex insecta* has not yet been resolved. In-depth genome analysis will help shed light on the possibility of its independent existence outside host cells and which key medium additives are crucial for growth. The preliminary draft genome of the *Sodalis* associated with *B. trigonica*, described in this study, reveals a heavily reduced genome size of 1.57 Mb compared to eight other *Sodalis* genomes isolated from different insect species (with a genome size range from 1.3 to 5.1 Mb). This truncated genome might be an indication of a higher dependency on its insect host, thus limiting its cell-free existence, as *Sodalis* species that have been cultured in vitro tend to have larger genomes (> 4 MB). 

Accession numbers of sequences generated in this study: 

*Sodalis* draft genome - GCA_003668825.1*Sodalis* 16S rDNA - MH973240, MH973241*Sodalis* partial GroEL - MH987777*Spiroplasma* 16S rDNA - MH973257CLso 16S rDNA - MH986752

## Figures and Tables

**Figure 1 microorganisms-08-00692-f001:**
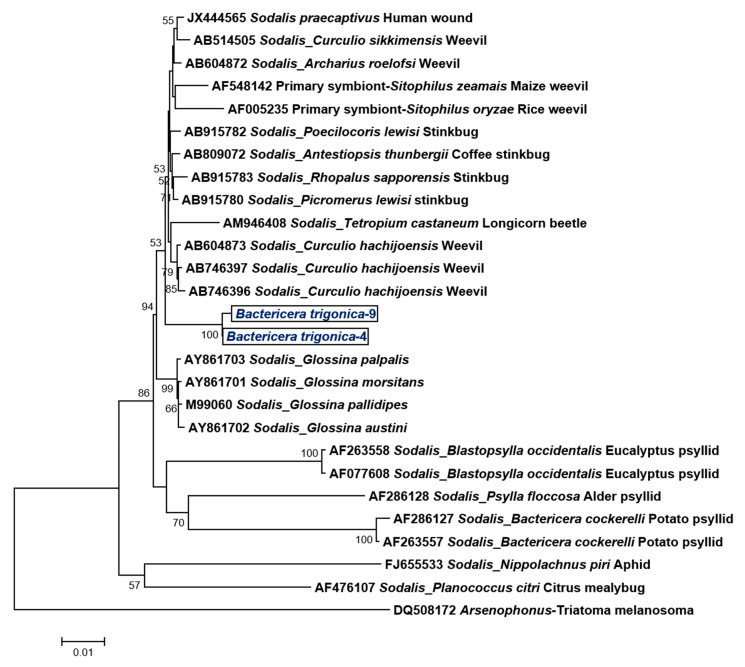
Maximum likelihood phylogenetic tree of nucleotide sequences of *Sodalis*-like 16S rDNA (1440 bp) from diverse insects by HKY + G + I nucleotide substitution model using MEGA 6. The *Sodalis* members identified inside *B. trigonica* are indicated inside a text box.

**Figure 2 microorganisms-08-00692-f002:**
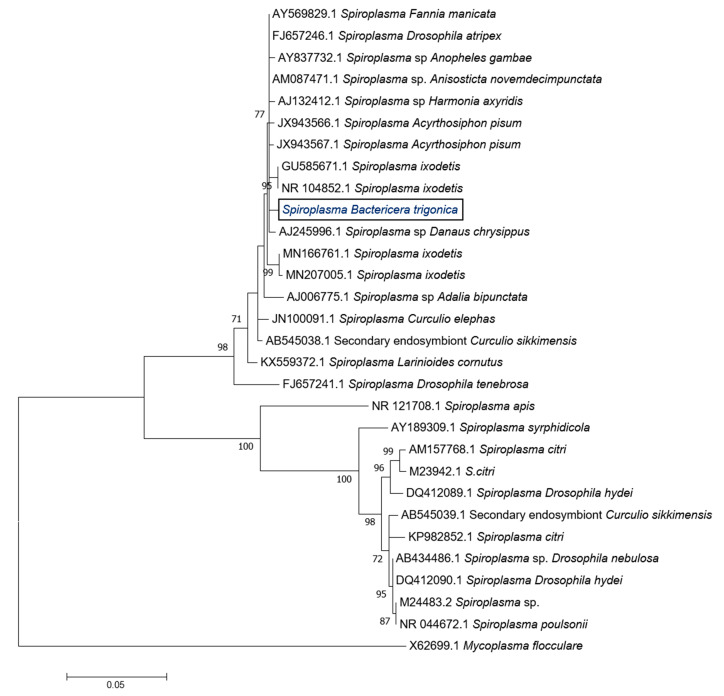
Maximum likelihood phylogenetic tree of 16S rDNA nucleotide sequences of *Spiroplasma* (1193 bp) by T93 + G nucleotide substitution model using MEGA 6.

**Figure 3 microorganisms-08-00692-f003:**
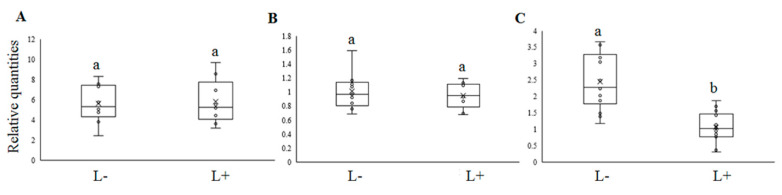
Relative quantities of *Sodalis* (**A**), *Spiroplasma* (**B**) and BtDNV (**C**) inside CLso-infected (L+) and uninfected (L-) psyllids. Different lowercase letters indicate significant differences of means (*p* < 0.001).

**Figure 4 microorganisms-08-00692-f004:**
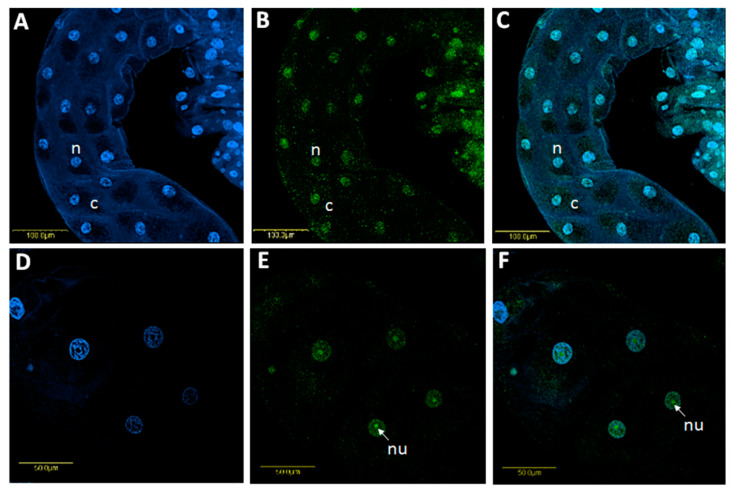
Localization of *Sodalis* in *B. trigonica* midgut using fluorescent in situ hybridization (FISH) with a *Sodalis-*specific probe. **A** and **D**: DAPI staining of BT midgut nuclei (blue). **B** and **E**: Localization of *Sodalis* (green) inside *B. trigonica* midguts by FISH, mostly showing specific localization inside the nucleus and the nucleolus and some localization in the cytoplasm around the nucleus. **C** and **F**: Overlay of *Sodalis* and DAPI staining inside midgut nuclei under a bright field. Arrows indicate, n: nucleus; nu: nucleolus; c: cytoplasm.

**Figure 5 microorganisms-08-00692-f005:**
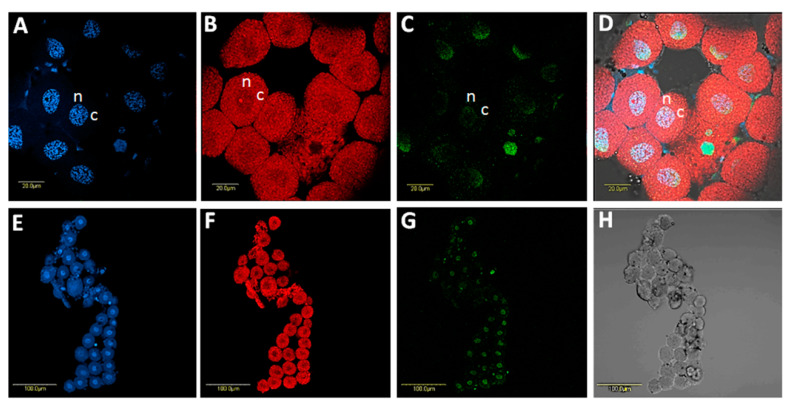
Localization of *Sodalis* in *B. trigonica* bacteriocytes using FISH with *Sodalis-*specific probe. **A**, DAPI-stained bacteriocyte nuclei. **B**, Same photo as in A showing the localization of *Carsonella,* the primary symbiont of psyllids (red) in the bacteriocytes. **C**, *Sodalis* localization (green) inside bacteriocyte nuclei. **D**, Overlay of A, B and C under a bright field. E-G show a cluster of bacteriocytes with DAPI, *Carsonella* and *Sodalis* staining, respectively. H shows the same cluster of bacteriocytes under a bright field. n: nucleus; c: cell cytoplasm.

**Figure 6 microorganisms-08-00692-f006:**
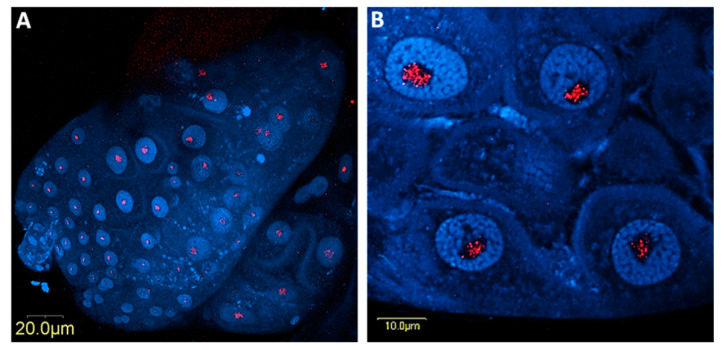
Immunolocalization of *Sodalis* (red) in *B. trigonica* midguts using anti-*Sodalis* monoclonal antibodies and cy3 labelled anti-mouse secondary antibody. The midgut nuclei were stained with DAPI (blue). **A**, the posterior midgut and **B**, in higher magnification showing four nuclei with *Sodalis* localized inside (red).

**Figure 7 microorganisms-08-00692-f007:**
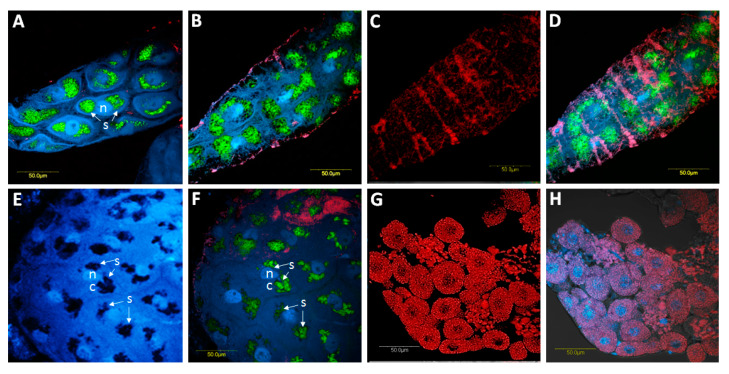
Co-localization of *Spiroplasma* and CLso in the *B. trigonica* midgut using FISH. **A**, **B**, focal plans showing the localization of *Spiroplasma* (green) in BT midguts by FISH and DAPI-stained nuclei (blue). In these focal planes, some localization of CLso appears (red). **C**, the same portion of the midgut in B showing the localization of CLso (red). **D**, overlay of *Spiroplasma*, midgut nuclei and CLso under bright field. E-F, other views of *Spiroplasma* localization (green) and DAPI-stained nuclei (blue). **E**, *Spiroplasma* patches around the nuclei without any staining. **F**, *Spiroplasma* patches with staining. **G**, FISH using *Carsonella* (red) and *Spiroplasma* probes in bacteriocytes. *Spiroplasma* was not detected inside bacteriocyte cells. **H** is the overlay of *Carsonella*, *Spiroplasma* and DAPI-stained nuclei in bacteriocytes. n: nucleus; c: cellular cytoplasm. Arrows indicate, s: *Spiroplasma*.

**Figure 8 microorganisms-08-00692-f008:**
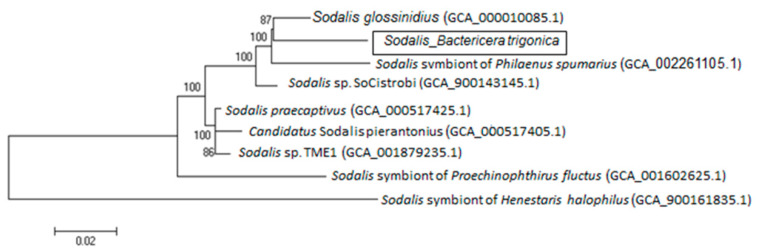
Phylogeny of 3404 orthologous genes (concatenated sequences of 347,000 bp each) of the *Sodalis* sp genomes available from the GenBank including the *Sodalis* identified from *B. trigonica*.

**Table 1 microorganisms-08-00692-t001:** Primer and probe sequences used in this study. CLso: *Candidatus* Liberibacter solanacearum.

Primer	Target	Sequence (5′-3′)	Product Size
Actin-F Actin-R	*B. trigonica* actin gene	AGATGACCCAGATCATGTTTGA AGGGCGTAACCTTCATAGATG	160 bp
Lso-F Lso-R	CLso omp A gene	CCATATCCAAATTTCAAAGAACC ATGCCACGTGAAGGTTTGAT	152 bp
Sod-F1 Sod-qR2	*Sodalis* GroEL gene	CCAAAGACGGCGTATCAGTT GCCTTCGTTGACGATAGACT	160 bp
Spir-F Spir-R	*Spiroplasma* 16S rDNA	CTGCCTCATGGCAACACTTA TTTCATGTGTAGCGGTGGAA	170 bp
Denso-qF Denso-qR	BtDNV VP4 gene	CACCGAGAACACGCACTTTG GACCAAAACTCTGGAGGGCA	149 bp
**Probe**	**Target**	**Sequence (5′** -**3′)**	**Reference**
*Carsonella*	16S rDNA	Cy3-CGCGACATAGCTGGATCAAG	[31] (pB-1664)
CLso	16S rDNA	Cy3-GCCTCGCGACTTCGCAACCAAT	This study
*Sodalis*	16S rDNA	Cy3/Cy5-GTTACCCGCAGAAGAAGCAC	This study
*Spiroplasma*	16S rDNA	Cy5-TTTCATGTGTAGGGGTGGAA	This study

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
