# Peer review of "An Intranuclear Sodalis-Like Symbiont and Spiroplasma Coinfect the Carrot Psyllid, Bactericera trigonica (Hemiptera, Psylloidea)"

_microorganisms, 2020, doi:10.3390/microorganisms8050692_

Round 1

Reviewer 1 Report

In this manuscript, authors investigate two symbionts of the carrot psyllid. I believe the results of the study are essential findings. However, many points in the MS are unclear; a lot of work is still required on the manuscript.

Introduction

1st paragraph. The concept of P-, S-symbionts is outdated. Sometimes it appears in modern publications in case the biology of analysed symbionts drastically differ, and it is convenient to consider them by P-, S- categories; 2) authors take old paradigm.

2nd paragraph has strange structure; telegraph style. Authors stat with psyllids, then Carsonella ruddii, Bactericera trigonica, and carrot. What idea of this part?

Lines 47-48. Carsonella ruddii is not closely related to Wolbachia, Rickettsia. It is gamma-proteobacteria.

3d paragraph. Line 58. ‘persistent manner’ ???

64-65 lines, what do authors use under the term ‘parasite’. Probably ‘symbiont’?

67-71 and 72-77 lines seems to have to be combined. Alternatively, remake it: objectives and a design of the study.

Results

The only two bacteria were found in RFLP, and DGGE analyses or just two were identified? No technical description of results; how many bands were analysed, how many clones were sequenced. What about Wolbachia, Carsonella, CLso, and facultative bacteria? Did the authors analyse the prevalence of CLso? How did authors detected densovirus? Also, I cannot find such information in the Material and Methods section.

168-169 lines and figure 1. Insufficient description in the MS and the figure. Where is evidence of Sodalis-like bacteria in B. trigonica? The only one isolate of Sodalis is presented on the figure. Where is a cluster of Sodalis bacteria, how authors defined it?

Lines 174-182. What is the idea of this paragraph? Reconstruct.

Also here. The whole-genome about two tens of Spiroplasma species are sequenced. Authors used only the S. citri group and two representatives of ixodetis and apis clades.

Lines 186-190. What is the motivation of this analysis, please briefly indicate it in the MS. Clarify why is CLso take a central role. Pools contained three females and two males. It seems it would be better to analyse males and females separately or individually. Explain it in the MS.

Figure 8. What were orthologous genes used? What was the length of concatenated sequences? Where is ‘Circular representation of the genome…’?

Discussion.

I doubt that the importance of carrot for Israel should be the first sentence in the Discussion section. Also, the first paragraph (255-280 lines) have considered everything. What the main idea of the first paragraph? Reconstruct.

Line 281. I do not find evidence that the isolate of Spiroplasma ‘is identical to S. ixodetes’. Moreover, there is no article about NR104852. Add other isolates of S. ixodetes for the phylogenetic analysis.

Lines281-282. ‘…Spiroplasma ixodetes, which is a species known to induce male killing in butterfly and ladybird beetle larvae…’ There are no such data, but there are data that closely related to S. ixodetes strains or Spiroplasma species can induce MK.

The authors revealed Spiroplasma only in the midgut tissues. We know that some lineages of Spiroplasma are transmitted vertically and some horizontally, via a plant host. Where does Spiroplasma come from? From carrot? According to phylogenetic analysis, we are dealing with a vertically transmitted lineage.

Lines 300-301. Is Schulz and Horn (2015) the first study of Rickettsia and Orientia intranuclear symbiosis?

Author Response

Comment 1: The concept of primary and secondary symbionts is outdated.

Response: The terms primary and secondary symbionts have been avoided and maintained as obligate and facultative symbionts.

Comment 2: 2nd paragraph has strange structure, authors start with psyllids, then carsonella, then B. trigonica and then carrots. Why?

Response: This structure was drawn to relate the endosymbionts of psyllids with the role of B. trigonica as a vector for CLas to the readers.

Comment 3: Carsonella is not closely related to Wolbachia or Rickettsia.

Response: This sentence was meant as secondary symbionts inside psyllids closely related to Arsenophonus,….Wolbachia. However ‘closely related’ was replaced with ‘such as’ (line 48) to avoid confusion.

Comment 4: Persistent manner?

Response: The persistent and circulative manner is used for insect transmitted pathogens which are circulate through the midguts of the insect vector to reach the salivary glands for transmission and such pathogens also persist inside the vector for life time. This is a widely used terminology do describe pathogen-vector interactions.

Comment 5: Why authors use the term parasites for Wolbachia in mosquitoes?

Response: The word parasite has been removed.

Comment 6: Combine line 67-77.

Response: The lines have been combined (line 68-75).

Comment 7: Technical descriptions of RFLP are missing.

Response: All details with number of colonies screened, number of colonies with unique RFLP patterns sequenced have been added to the methods section now (line 102-105).

Comment 8: what about wolbachia, carsonella, CLso and densovirus?

Response: It is mentioned in the text that only Sodalis, Spiroplasma and Clso were detected by RFLP (line 171-72). It is also discussed about the absence of Arsenophonus and Wolbachia in B. trigonica (line 290) while they have been detected from other psyllids using similar methods. Densovirus was detected in a previous study of ours (Reference 26) and we have assessed its prevalence inside B. trigonica in this study.

Comment 9: Insufficient description in Fig.1.

Response: The description has been modified.

Comment 10: Where is the evidence of Sodalis like bacteria in B. trigonica? Why only 1 isolate is presented in the figure?

Response: The phylogenetic tree presented in this study is made from 16S rDNA sequences of Sodalis like symbionts from diverse insects. Our isolates cluster closely with other Sodalis like sequences from weevils and stink bugs. The accession numbers of our isolates have also been mentioned at the end of the manuscript. Two isolates are presented in this figure. Other three sequences analysed were identical to these.

Comment 11: Reconstruct paragraph (lines 174-82)

Response: Paragraph split into two and sentences added for better clarity (lines 180-90).

Comment 12: Authors used only 2 representatives of the ixodetes and apis group Spiroplasma.

Response: The NCBI reference sequence for S. ixodetes Y32 strain (NR_104852.1) was used in the phylogeny, which was 99.3% similar to the Spiroplasma from B. trigonica. However, the tree has been re-drawn with more S. ixodetes isolates including one from ATCC:3835 (GU585671).

Comment 13: Mention the motivation for quantifying symbionts with CLso infection status.

Response: A sentence has been added to justify the need for this (line 195-97).

Comment 14: What were orthologous genes used? What was the length of concatenated sequences?

Response: A set of 3404 orthologous genes identified were used to make the tree. The total length of concatenated sequences (347,000 bp) is included in the figures now.

Comment 15: Reconstruct the first paragraph of the discussion.

Response: First paragraph of the discussion has been removed as per reviewer’s suggestion.

Comment 16: No clear evidence that this spiroplasma is identical to S. ixodetis. Include more Spiroplasma ixodeted for the phylogeny.

Response: The Spiroplasma from B. trigonica is 99.3% identical to the S. ixodetis Y32 strain which had been used for the phylogeny. NR104852 is the NCBI reference sequence for S. ixodetis Y32 strain. However, more three sequences have been included in the phylogenetic tree now.

Comment 17: Closely related species of S. ixodetis can induce male killing.

Response: The change is made and the word closely related is included.

Comment 18: Where does the Spiroplasma come from?

Response: As suggested by the reviewer, the Spiroplasma is transferred vertically from females to offspirings.

Comment 19: Is Schultz and Horn the first study of intranuclear Rickettsia and Orientia?

Response: The evidence of Rickettsia and Orientia have been taken from a review article by Schultz and Horn.

Reviewer 2 Report

This work by Ghosh et al. provides an interesting first characterization of Sodalis-like and Spiroplasma endosymbionts in an important agricultural pest (carrot psyllid), as well as additional characterization of a previously discovered densovirus (BtDNV) infection in the same host. The approaches used include characterization of infection prevalence in field populations, relative quantitation of titer in laboratory populations, as well as phylogenetic analysis of the novel endosymbionts and FISH and immunohistochemistry to determine their cellular localization in tissues.

Overall, this paper is written clearly, employs appropriate methods, and presents intriguing novel findings such as the intranuclear localization of Sodalis and interesting phylogenetic placement of the endosymbionts. It was an exciting read and contains some beautifully captured images. I do not have major concerns regarding the science in the manuscript as the main conclusions of the authors are appropriately supported. Nonetheless, I do feel that some aspects of the manuscript could be improved prior to publication. In particular, some of the methods are somewhat unclear and slightly limit interpretation of the data. In addition, some of the text in the results and discussion section could be amended to more closely reflect the figures and findings.

Some specific comments for the authors:

ABSTRACT

Line 20: It would be useful to define acronyms in the abstract

Line 30: “2-fold” not “2-folds”, this also applies throughout the manuscript

It would be useful to include a concluding statement in the abstract regarding the novelty and implications of the findings.

INTRODUCTION

Line 39: As the authors indicate, there is a plethora of evidence that primary endosymbionts share metabolic pathways with their hosts. However, no references are given for this statement. It would be useful to provide a few key examples.

MATERIALS AND METHODS

Line 80: If PCR was used to determine CLso infection status, then the process is destructive. Does CLso reach fixation in populations? If so, then one can assume from PCR testing of sub-samples that a lab population is stably infected or uninfected, but otherwise not. Is the fixation achieved by rearing on infected celery? This is likely due to my lack of familiarity with rearing this species, but some editing/clarification in this section would be beneficial to better express how the authors maintained separate populations with 0% and 100% infection rates.

Line 113: Were individuals collected from the same or different fields? How many different locations are represented in the sampling? With only 40 individuals, some of the conclusions of the authors regarding fixation in the population are weak, especially if only a few locations were sampled.

Line 125: The authors state that total DNA was extracted from a pool of 5 insects, which included 3 females and 2 males. However, no information is given on replicates. I assume based on the statistics conducted that more than just 1 pool of CLso infected and 1 pool of uninfected were compared. Please provide replication information.

Line 125: In addition, why were males and females pooled together? It would have been more informative to analyze the sexes separately, as it is known that the titer of some endosymbionts can vary between sexes. Do the authors have a particular justification for this?

Line 141: Though it is entirely appropriate to cite previous methods used by the authors, it would be convenient for readers to provide at least a brief statement to describe these previous methods here and in the section immediately following.

Line 152: It is important to provide additional information on trimming parameters.

RESULTS

Line 195-196: While I agree with the authors that Sodalis localization is in the nucleus and nucleolus, there does seem to be quite a bit of cytoplasmic localization based on Figure 4 FISH as well, but this is not mentioned in the text. It would be appropriate to revise this statement to better align with Figure 4 and Figure 6 (i.e. localization is in the nucleus/nucleolus as well as the cytoplasm based on FISH, but the latter is not detected by immunostaining).

DISCUSSION

The discussion is generally strong and addresses several important points that were raised in my mind, such as speculation on where the Sodalis may have been acquired from. The discussion of the implications of intranuclear localization is particularly interesting in my opinion. My main concern pertains to the use of the term “fixation” when referring to detection of Sodalis and densovirus in 40/40 of individuals tested from the field (e.g. Line 270). This is done in the discussion as well as throughout the manuscript. However, given the small sample size tested, I think it would be more accurate to state that prevalence was 100% in the samples tested, without making sweeping conclusions at the population level, since 40 individuals is hardly representative of a single population, let along multiple populations.  

Author Response

Comment 1: Define acronyms in the abstract.

Response: We agree with the reviewer, however full expansion of the acronyms was not possible due to maximum word limits (200 words). However, both the acronyms have been expanded in the methods section for reader’s convenience (line 86, 102-03).

Comment 2: Convert two folds to fold throughout the manuscript.

Response: The change was made (line 31, 190).

Comment 3: Include a concluding statement in the abstract.

Response: A concluding sentence has been added (line 29-30).

Comment 4: Endosymbionts share metabolic pathways: no references have been cited.

Response: A review which details such collaboration has now been cited (line 38).

Comment 5: How authors maintain CLso uninfected and infected psyllids?

Response: The infection status of psyllid population was determined by qPCR detection of celery plants after 2 months from release of insect (line added 83-85). Colonies identified positive or negative are maintained on separate cages and reared on infected/uninfected celery plants. Additions were made to the methods section for clarification of readers.

Comment 6: Were individuals collected from different fields.

Response: Yes the psyllid samples were collected from different fields and have been mentioned in the text (line 116-117).

Comment 7: With limited sample numbers it is inconclusive whether symbionts are fixed in a population.

Response: Throughout the manuscript the term ‘fixed’ has been replaced as ‘detected in 100% of samples tested’ for Sodalis or BtDNV.

Comment 8: No information on number of replicates of samples used for qPCR.

Response: The exact replicate numbers of CLso infected and uninfected samples used for qPCR has now been added (Line 131-32).

Comment 9: Why were males and females pooled together for the qPCR.

Response: We previously show that the symbionts infect both male and female psyllids and thus combined them in same numbers to relate the dynamics of symbionts irrespective of the sexes of the psyllids.

Comment 10: Brief description of the methods used for FISH.

Response: The method used for FISH has been briefly described.

Comment: Additional information on trimming of reads

Response: The software used for trimming of reads have been added now (line 161-62).

Comment 11: Cytoplasmic localisation of Sodalis inside the midgut cell by FISH.

Response: The cytoplasmic localisation has now been mentioned in the text (202-203).

Comment 12: The term fixation of insects is not suitable with respect to the limited sample size.

Response: The term fixation has been removed throughout the manuscript and replaced by ‘detected in 100% of the samples tested’.

Round 2

Reviewer 1 Report

The authors specified many issues in the new version of MS. However remains unclear how B. trigonica catch the Spiroplasma infection.

Other comments:

38-42 lines. Some obligate symbionts synthesize B vitamins (not only essential amino acids), and some obligate symbionts are not located in a bacteriocyte.

50 line. Need a sentence that introducing the symbionts in psyllids before «Ð¡andidatus Carsonella…».

52 line. Delete “(B.trigonica)”

56-57 lines. Omit ‘… a popular vegetable grown in many European countries.’

56-63 lines form a separate paragraph

70 and further. Aims and results paragraph.

88 line. B. trigonica

97-98 lines. Too long title. Probably it is ‘DNA extraction and symbiont screening’. Also, 2.3 is added. Or it should be subdivided into experimental procedures. 2.2. DNA extractions; 2.3. PCR…

108, 117-118 lines. Add 5’-, -3’.

126 line. Seems ‘and BtDNV’ should be added after ‘symbionts’, and then delete 128-130 lines.

126-128 lines. About “amplifying the actin gene”. Is it used for the estimation of the quality of DNA samples? If so it should be in the first paragraph of M&M.

135 line. ‘5 insects (3 female and 2 male…’ it was not explained in author responses.

142 line.

181-183 lines. How RFLP and DGGE analysis identified bacterial genera without sequencing and following BLAST tool search?

Figure 1. Where are the boundaries of Sodalis-like bacteria? Here the mix of host names and bacterial names (in particular Sodalis praecaptivus is a bacterial name, but other). Make clear it on the figure. Add ‘isolate-9’ and ‘isolate-4’ and GenBank numbers to your samples.

Figure 2. The genera should be italicized.

307 line. Again, it is not “identical” but sure (308 line) “close related”.

309-310 lines. What is sex ratio in your collection?

310-311 lines. Please consider a possible way of Spiroplasma acquisition.

320 line. Unclear ‘…an infectious clone of the BtDNV…’

348 line. The portion after ‘psyllids’ may be omitted.

Author Response

Comment 1: Some obligate symbionts can synthesize vitamins.

Response: This has been added (line 39).

Comment 2: Some obligate symbionts are not located in bacteriocytes?

Response: The sentence has been rephrased as ‘mostly located (line 36)’.

Comment 3: Add a sentence to introduce the symbionts.

Response: The sentence has been divided into two to separate Carsonella and also additional details have been added (lines 47-51).

Comment 4: Delete B. trigonica.

Response: The B. trigonica acronym is being retained for easy understanding of the readers.

Comment 5: Delete ‘popular vegetable grown in Israel and Europe.

Response: Deleted (line 55).

Comment 6: Line 56-63- separate paragraph.

Response: The paragraphs have now been separated.

Comment 7: Aims and results-separate paragraph.

Response: Paragraphs have been separated.

Comment 8: B. trigonica (line 88)

Response: Abbreviated as suggested.

Comment 9: Section 2.2 heading is too long.

Response: The heading has been shortened and modified.

Comment 10: Add 5’ and 3’ to the primer sequence.

Response: Added as suggested.

Comment 11: Add BtDNV to line 126 and delete line 128.

Response: The modifications are now made as suggested.

Comment 12: The actin gene amplified as internal control should go to the first paragraph of M&M.

Response: The actin gene was amplified to verify the quality of each psyllid sample tested for symbionts and was done simultaneously with the symbiont screening. Thus we think that it would be easier for readers to comprehend that quality of each DNA sample was tested. Hence we retain the sentence there.

Comment 13: 5 insects (3 females and 2 males) were not included in author response.

Response: This was a common query from both Reviewer 1 and 2.  This has been responded against comment 9 of the 2nd reviewer, previously. However we provide the response again here for your reference.

‘We previously show that the symbionts infect both male and female psyllids and thus combined them in same numbers to relate the dynamics of symbionts irrespective of the sexes of the psyllids.’

Comment 14: How DGGE and RFLP identified bacteria without sequencing?

Response: The section has been modified for better clarity of readers. The sequencing of clones have now been mentioned (170-72).

Comment 15: what are the boundaries of Sodalis like bacteria? Why mix of host and bacterial names?

Response: Only sequence of Sodalis like bacteria have been included except an out group (Arsenophonus) for the phylogenetic tree. Majority of the sequences used from genbank are already published as Sodalis spp. And references of them are cited in the discussion section. However, Sodalis has been added to each of them now.

Also the Sodalis identified from the particular insect species (aphids, mealybugs, stink bugs etc.) along with their latin names have been used in this tree for reader’s convenience.

Comment 16: Provide genbank numbers on the tree.

Response: All genbank numbers of the identified bacteria have been provided at the end of the manuscript. This is to separate clones identified in this study in the phylogenetic tree.

Comment 17: The genera should be italicized (Fig.2).

Response: The change is made as suggested.

Comment 18: Change identical to closely related (line 307)

Response: Change made (line 285).

Comment 19: What is the sex ratio in your collection?

Response: The sex ratio has not been estimated in the lab population yet.

Comment 20: Please consider a way for Spiroplasma acquisition.

Response: A sentence has been added.

Comment 21: Unclear how infectious clones of btDNV can help understand interactions.

Response: The sentence has been removed.